# Laminin Receptor-Mediated Nanoparticle Uptake by Tumor Cells: Interplay of Epigallocatechin Gallate and Magnetic Force at Nano–Bio Interface

**DOI:** 10.3390/pharmaceutics14081523

**Published:** 2022-07-22

**Authors:** Sheng-Chieh Hsu, Nian-Ping Wu, Yi-Ching Lu, Yunn-Hwa Ma

**Affiliations:** 1Department of Biomedical Sciences, College of Medicine, Chang Gung University, Guishan, Taoyuan 33302, Taiwan; schsu@mail.cgu.edu.tw; 2Master Program in Biotechnology Industry, College of Medicine, Chang Gung University, Guishan, Taoyuan 33302, Taiwan; 3Graduate Institute of Biomedical Sciences, College of Medicine, Chang Gung University, Guishan, Taoyuan 33302, Taiwan; ap82021@gmail.com; 4Department of Physiology and Pharmacology, College of Medicine, Chang Gung University, Guishan, Taoyuan 33302, Taiwan; yichinglu@gmail.com; 5Department of Medical Imaging and Intervention, Chang Gung Memorial Hospital, Guishan, Taoyuan 33305, Taiwan

**Keywords:** magnetic nanoparticles, endocytosis, epigallocatechin gallate, 67LR, Akt

## Abstract

Epigallocatechin gallate (EGCG), a major tea catechin, enhances cellular uptake of magnetic nanoparticles (MNPs), but the mechanism remains unclear. Since EGCG may interact with the 67-kDa laminin receptor (67LR) and epidermal growth factor receptor (EGFR), we investigate whether a receptor and its downstream signaling may mediate EGCG’s enhancement effects on nanoparticle uptake. As measured using a colorimetric iron assay, EGCG induced a concentration-dependent enhancement effect of MNP internalization by LN-229 glioma cells, which was synergistically enhanced by the application of a magnetic field. Transmission electron microscopy demonstrated that EGCG increased the number, but not the size, of internalized vesicles, whereas EGCG and the magnet synergistically increased the size of vesicles. EGCG appears to enhance particle–particle interaction and thus aggregation following a 5-min magnet application. An antibody against 67LR, knockdown of 67LR, and a 67LR peptide (amino acid 161–170 of 67LR) attenuated EGCG-induced MNP uptake by 35%, 100%, and 45%, respectively, suggesting a crucial role of 67LR in the effects of EGCG. Heparin, the 67LR-binding glycosaminoglycan, attenuated EGCG-induced MNP uptake in the absence, but not presence, of the magnet. Such enhancement effects of EGCG were attenuated by LY294002 (a phosphoinositide 3-kinase inhibitor) and Akt inhibitor, but not by agents affecting cGMP levels, suggesting potential involvement of signaling downstream of 67LR. In contrast, the antibody against EGFR exerted no effect on EGCG-enhanced internalization. These results suggest that 67LR may be potentially amenable to tumor-targeted therapeutics.

## 1. Introduction

Nanoparticles may act as drug carriers to overcome cellular barriers and facilitate delivery of drugs or therapeutic agents for cancer therapy [1,2,3]. In targeted delivery, different nanoparticles, drugs, and imaging agents may be immobilized to or encapsulated within biocompatible organic/inorganic shell structures to form a multifunctional system. Nanoparticles can passively target tumor tissues because of the large gaps in the endovascular wall of the tumor, as the enhanced permeability and retention effect [4]. Subsequently, the interaction between drug-loaded nanoparticles and tumor cells is often the key to cellular internalization, which may be crucial to the efficacy of chemotherapeutic drugs [5,6,7].

Magnetic nanoparticles (MNPs) with an iron oxide core exhibit great potential to serve as carriers for targeted delivery of drugs due to their superparamagnetic properties, which allow MNPs to be guided by magnetic force against blood flow to achieve targeting effects, and back to a non-magnetic state after removal of the magnetic field [8]. In addition to drug delivery, MNPs may be applied to gene transfection, magnetic separation, magnetic resonance imaging, and magnetic hyperthermia. Nanoparticle clusters (>200 nm in size) are generally taken up by cells via micropinocytosis, and the surface coating of nanoparticles may determine the molecular mechanisms of the endocytosis process [9]. Drug targeting systems driven by receptor-mediated endocytotic pathways exhibit potential in theranostics since several specific plasma membrane biomarkers that entrain endocytosis have been identified for tumor therapy. Membrane receptors that are potential targets in drug delivery include epidermal growth receptor (EGFR), folate receptors, integrin receptors, and transferrin receptors [10,11]. Nevertheless, many parameters may affect the nanoparticle internalization by cells, including the size, shape, and surface chemistry of the nanoparticles, as well as cellular factors such as the cell type, membrane tension, and receptor [12,13].

We have previously demonstrated enhancing effects of tea catechins on nanoparticle internalization by glioma and other cells [14,15,16]. The major polyphenols of tea catechins are catechin, gallocatechin, (−)-epicatechin, (−)-epigallocatechin, (−)-epicatechin-3-gallate, and (−)-epigallocatechin-3-gallate (EGCG). EGCG is the most abundant component among these catechins, which have been demonstrated to exert antioxidant, anti-inflammatory, anti-atherosclerosis, pro-apoptosis, neuron-protective, and anti-carcinogenic effects [17,18,19,20]. Although EGCG exhibits various biological and pharmacological properties, the mechanism of EGCG-enhanced internalization of MNPs has not been elucidated. In addition, a synergistic effect of EGCG and magnetic force on MNP uptake by a variety of cells has been reproducibly demonstrated without a known mechanism [14,15,16].

Tachibana et al. identified the 67-kDa laminin receptor (67LR) as a cell-surface EGCG receptor using a subtraction cloning strategy [21]. The 67LR is a non-integrin laminin receptor that is associated with growth, metastasis, and drug resistance and may be responsible for EGCG-induced effects on cancer cells [22,23,24]. It binds to YIGSR (Tyr-Ile-Gly-Ser-Arg) of the laminin β1 chain, which helps cells adhere to the basement membrane. The peptide YIGSR competes with laminin for the binding of integrin or 67LR [25,26]. The EGCG–67LR interaction site was identified as the sequence 161–170 of 67LR using a surface plasmon resonance biosensor [27]. The amino acid sequences 161–180 (peptide G) and 205–229 of 67LR bind to laminin in a heparin-dependent and independent manner, respectively [28]. There are two forms of 67LR, 37-kDa and 67-kDa; it is generally accepted that the 37-kDa protein is a precursor of the 67-kDa protein, with homo- or hetero-dimerization of the 37-kDa protein by fatty acid acylation [29]. The 37-kDa precursor has been identified as a ribosomal protein that belongs to the 40S ribosomal subunit, which interacts with histones H2A, H2B, and H4 of DNA and participates in translation [30].

Many signaling cascades triggered by EGCG–67LR interaction have been explored for the selective anti-tumor activity of EGCG [31]. The 67LR has been shown to express on cell membrane microdomains known as lipid rafts, which are enriched in sphingomyelins, glycosphingolipids, and cholesterol [32]. Interestingly, the downstream signaling pathways of EGCG–67LR have been demonstrated to alter the structure/function of lipid rafts and thus modulate the endocytosis process; the interaction of EGCG and 67LR induced activation of the protein kinase B (PKB; Akt)/endothelial nitric oxide synthase (eNOS)/ nitric oxide (NO)/ soluble guanylyl cyclase (sGC)/ cyclic guanosine monophosphate (cGMP) pathway, which has been shown to increase membrane fluidity and apoptosis in several cancer cell models [24]. Furthermore, EGCG has been reported to activate myosin phosphatase and reduce phosphorylation of the myosin regulatory light chain through the 67LR signaling pathway, which induces actin cytoskeleton remodeling to affect endocytosis [33]. Such effects may facilitate intracellular delivery of nanocarriers with therapeutic agents. Nanoparticles with immobilized EGCG have been shown to exhibit enhanced uptake by tumor cells [34,35] and enhanced therapeutic efficacy in tumor models [13,34,35,36,37]. Limited evidence indicates that the 67LR antibody attenuates the enhancement effects of EGCG on cellular uptake of EGCG nanocomposites [34,36]. However, it remains unknown whether a receptor on the plasma membrane may mediate the enhancing effect of EGCG in the milieu on cellular uptake of nanoparticles.

In addition to 67LR-mediated signaling pathways, EGCG may suppress specific receptor tyrosine kinases and related downstream signaling pathways [38]. Epidermal growth factor receptor (EGFR) is a transmembrane protein that is primarily expressed on the cell membrane structure lipid raft. Upon activation by its specific ligands, EGFR changes conformation from an inactive monomeric form to an active dimer [39]. Phosphorylation at tyrosine triggers downstream signaling pathways and induces endocytosis of EGFR. A previous study reported that EGCG alters plasma membrane organization and causes internalization of EGFR, resulting in the suppression of colon cancer cell growth [40,41]. Another study demonstrated that phosphorylation of EGFR at serine 1046/1047 via activation of p38 mitogen-activated protein kinases played a pivotal role in EGCG-induced downregulation of EGFR in colon cancer cells [42]. However, it is not known whether EGFR is involved in EGCG-induced nanoparticle internalization.

In this study, we investigate whether a receptor and its downstream signaling molecules may mediate the EGCG-induced augment of MNP internalization by tumor cells with and without the influence of a magnetic field. We demonstrate the important role of 67LR and PI3K/Akt activation, but not EGFR, in EGCG-enhanced MNP internalization, which provides a novel molecular mechanism for chemical and magnetic synergy at the nano–bio interface.

## 2. Materials and Methods

### 2.1. Materials

The magnetic nanoparticle nanomag^®^-D COOH (dextran-MNP with Fe_3_O_4_ core; 250 nm) was purchased from Micromod Partikeltechnologie GmbH (Rostock, Germany). FluidMAG-CMX (CMX-MNP; 200 nm) was purchased from Chemicell GmbH (Berlin, Germany). Dulbecco’s modified Eagle’s medium (DMEM) with or without nutrient mixture F-12, 0.5% trypsin-EDTA, and antibiotic-antimycotic solution were purchased from Invitrogen (Carlsbad, CA, USA). Fetal bovine serum (FBS) was from Hyclone (Logan, UT, USA). Epigallocatechin gallate (EGCG), ammonium persulfate (APS), potassium thiocyanate (KSCN), NaCl, NaF, Na_3_VO_4_, phenylmethylsulfonyl fluoride, glutaraldehyde, paraformaldehyde (PFA), cacodylate buffer, and MISSION^®^ esiRNA (EHU109791) were purchased from Sigma-Aldrich (Saint Louis, MO, USA). EGFR antibody (C225), EGF, BAY41-2272, and zaprinast were purchased from Merck Millipore (Burlington, MA, USA). GAPDH antibody was purchased from GeneTex (Taipei, Taiwan). Antibodies for Akt (#4685), Akt pT308 (#2965), and Akt pS473 (#4060) were purchased from Cell Signaling (Danvers, MA, USA). 67LR antibody (MLuC5, ab133645) and the secondary antibody donkey anti-rabbit IgG H&L, Alexa Fluor^®^ 647 (ab150075) were purchased from Abcam (Cambridge, UK). Goat anti-rabbit IgG (H + L) secondary antibody (A11034), Silencer^®^ Select Negative Control siRNA #1 (Ambion #4390843), Lipofectamine^®^ RNAiMAX, and ProteoJET™ Membrane Protein Extraction Kit (#K0321) were purchased from Thermo Fisher (Waltham, MA, USA). The 67LR peptide (67LRp, amino acid 161–170 of 67LR, IPCNNKGAHS) was purchased from BIOTOOLS (Taipei, Taiwan). Protease/phosphatase inhibitors were purchased from GoalBio (Taipei, Taiwan). Desipramine was purchased from MedChem Express (Monmouth Junction, NJ, USA); 8-Br-cGMP was purchased from Enzo Life Sciences (Farmingdale, NY, USA). The NdFeB magnet was purchased from New Favor Industry Co. (Taipei, Taiwan).

### 2.2. Cell Culture

Human glioma cell lines (LN-229 and U87MG) and an epidermoid carcinoma cell line (A431) from Bioresource Collection and Research Center, Food Industry Research and Development Institute, Taiwan were maintained in DMEM or DMEM/F-12, containing 10% FBS and 1% penicillin/streptomycin/amphotericin B mixture in an incubator supplied with 5% CO_2_ at 37 °C. As described previously [14], culture cells in 24-well plates were subjected to magnetic force by placement on top of a homemade plate filled with a cylindrical NdFeB magnet with a magnetic field of 3.4 kGauss at the center of each well to facilitate MNP sedimentation on the surface of the cells for 5 min only, and defined as Mag−; meanwhile, Mag+ denotes application of the magnet for 2–24 h.

### 2.3. Transmission Electron Microscopy (TEM) and Image Analysis

After incubation with MNPs and EGCG for 24 h, U87MG cells were fixed with 3% glutaraldehyde and 2% paraformaldehyde (PFA) in cacodylate buffer (0.1 M, pH 7.4) at 4 °C for 2 h. Then, samples were post-fixed with 0.5% glutaraldehyde in cacodylate buffer at 4 °C and processed by Microscopy Core Laboratory of Chang Gung Memorial Hospital for TEM analysis. The size and number of vesicles per cell were obtained using image J (v1.51, National Institute of Health, Bethesda, Maryland).

### 2.4. 3D Cell Explorer

LN-229 cells were cultured in a 35-mm glass dish (α-plus) until 70% confluence, followed by incubation with MNPs (10 μg/mL) in the absence and presence of the EGCG (10 μM) for 2 h. The refractive index of cells and MNPs was acquired with a 3D Cell Explorer Nanolive microscope (Nanolive, Tolochenaz, Switzerland). A timelapse of cells was acquired with 4D live cell imaging.

### 2.5. MNP Uptake Assay

Cells were cultured in a 24- or 96- well culture plate until 90% confluence, followed by incubation with MNPs (50 μg/well) and EGCG in the absence and presence of the magnet for 2–4 h. Cell-associated MNPs (MNP_cell_) were determined using a colorimetric method as previously described [14]. Briefly, the cells were trypsinized and subjected to 10% HCl at 55 °C for 4 h. Ammonium persulphate (1 mg/mL) was added to convert ferrous to ferric ion, followed by addition of potassium thiocyanate (1 M) to allow formation of an iron–thiocynate complex. The amount of MNP_cell_ was determined at OD_490_ with a plate reader (BioTek Synergy HT, Winooski, VT, USA).

### 2.6. Protein Preparation and Extraction

After the required treatments, the cells were scraped in phosphate-buffered saline and collected in a microfuge tube followed by centrifugation at 3000 rpm for 5 min at 4 °C. Cells were lysed in lysis buffer (150 mM NaCl, 20 mM Tris-HCl pH 8.0, 1 mM EDTA, 1 mM phenylmethylsulfonyl fluoride, 10 mM NaF, 1 mM Na_3_VO_4_, 0.5% NP40, and protease/phosphatase inhibitors). Membrane and cytoplasmic proteins were obtained using the ProteoJET™ Membrane Protein Extraction Kit (Fermentas, Waltham, MA, USA) according to the manufacturer’s instructions.

### 2.7. Western Blot Analysis

Proteins in lysates were quantified using the Pierce™ BCA Protein Assay Kit (Thermo Fisher). Equal amounts of protein were fractionated by SDA-PAGE, transferred to a PVDF membrane, incubated with the appropriate primary antibody at 4 °C overnight, and then subjected to incubation with HRP-conjugated secondary antibodies at room temperature for 1 h. The bound antibodies were visualized by addition of enhanced chemiluminescence and exposure to an X-ray film.

### 2.8. Confocal Microscopy

LN-229 cells were seeded on the coverslips 24 h before the experiments. Lysotracker^®^ (1 μM) was used according to the manufacturer’s instructions for staining of lysosomes. After fixation with 2% paraformaldehyde (PFA), the cells were immunostained with anti-67LR antibody (Abcam #ab133645) and green fluorescent secondary antibody (Thermo Fisher #A11034), followed by counterstaining with DAPI for the nucleus. Images were acquired with an LSM 510 Meta laser confocal microscope system equipped with a 100×/1.4 oil immersion objective lens.

### 2.9. Knockdown of 67LR Using esiRNA

LN-229 (5 × 10^5^) cells were seeded in 6-cm culture dishes overnight before the experiments. To knock down 67LR, the cells were transfected with MISSION^®^ endoribonuclease-prepared siRNA (EHU109791) according to the manufacturer’s instructions. Silencer^®^ Select Negative Control No. 1 siRNA (Ambion #4390843) was used as a negative control. Transfection of esiRNA was conducted by using Lipofectamine^®^ RNAiMAX reagent according to the instruction manual. MNP_cell_ and 67LR/tubulin protein expressions were determined after transfection for 72 h.

### 2.10. Statistical Analysis

Values are expressed as means ± SEM. Results were analyzed by ANOVA followed by Duncan’s post hoc test when appropriate. Statistical significance was declared as *p* < 0.05.

## 3. Results

### 3.1. Synergetic Effects of EGCG and Magnetic Force on MNP Uptake

To determine the effects of EGCG and its interaction with magnetic force on MNP uptake, LN-229 and A431 cells were incubated with MNPs and EGCG for 2 h prior to an assay of cell-associated MNPs (MNP_cell_). Figure 1A illustrates an EGCG-induced concentration-dependent increase in MNP_cell_ with (Mag+) and without (Mag−) the magnet in LN-229 cells, and a synergistic effect of EGCG and the magnetic force, suggesting involvement of distinct mechanisms in augmenting MNP uptake. In A431 cells, EGCG and magnetic force exhibited similar synergetic effects, whereas EGCG alone did not increase MNP_cell_ (Figure 1B). In the Mag− group, a 5-min magnet was applied to facilitate MNP sedimentation, which induced MNP alignment in the culture medium (Figure 2A (left)). Incubation of EGCG with MNPs induced the formation of relatively longer fibrous aggregates in the absence (Figure 2A (right)) and presence (Appendix A) of LN-229 cells, suggesting an enhanced MNP–MNP interaction by EGCG upon 5-min magnet application. Similar results were observed in human U87MG cells (Figure 2B–D). EGCG greatly increased internalized MNPs in both the Mag− and Mag+ groups, as revealed by TEM (Figure 2B); EGCG did not alter the size of vesicles in the cytoplasm in the Mag− group, but significantly increased the vesicle size by more than 2-fold in the Mag+ group (Figure 2C). In addition, EGCG significantly increased vesicle number/cell by 4.9- vs. 1.4-fold in Mag− vs. Mag+ groups, respectively (Figure 2D). These results suggest that EGCG per se is associated with a more active endocytosis, whereas EGCG plus magnet may facilitate the formation of larger vesicles in the cytoplasm.

### 3.2. 67LR-Mediated MNP Uptake Induced by EGCG

To determine whether 67LR is present on the surface of LN-229 cells, confocal microscopy and western blot analysis were employed. Figure 3A illustrates fluorescent signal of 67LR in the cytoplasm and on the cell membrane, but not in the nuclei. Western blotting reveals that the expression of 67LR was primarily on the cell membrane vs. cytoplasm (Figure 3B), whereas tubulin and GAPDH were found to be in the cytoplasm section only, as expected. A 67LR-specific antibody (MLuc5) significantly attenuated effects of EGCG by up to 35% (Figure 3C), suggesting that EGCG-induced MNP uptake was mediated, at least in part, by 67LR. Since heparin binds to peptide G (sequence 161–180) of 67LR, which covers the potential EGCG binding sequence (161–170) [23], we examined whether heparin may attenuate the effects of EGCG on MNP uptake. In the absence of EGCG, heparin alone suppressed uptake of both CMX-MNPs (Figure 3D,E) and Dex-MNPs (Figure 3F), suggesting basal MNP_cell_ may be dependent on 67LR. In the Mag− group, heparin at 1 and 3 Unit/mL reduced the effects of EGCG on CMX-MNP uptake by 49% and 65% (Figure 3D), and on Dex-MNP uptake by 67% and 77%, respectively (Figure 3F). The concentration-dependent attenuation effect of heparin was demonstrated in the 0.01 to 1 Unit/mL range (Figure 3G). In addition, no significant cytotoxicity was detected by CCK-8 assay in response to heparin (up to 3 Unit/mL) for 24 h (Appendix A). These results suggest that EGCG may bind to the same region of the heparin-associated binding site on 67LR to exert the enhancement effect. However, heparin did not attenuate the enhancement effect of EGCG on MNP_cell_ with the application of magnetic force (Figure 3E), suggesting a different mechanism other than 67LR binding. Furthermore, a laminin-derived peptide YIGSR (Tyr-Ile-Gly-Ser-Arg) did not alter the effects of EGCG (Figure 3H), suggesting the binding site of the peptide YIGSR may be different from that of EGCG.

To further determine whether the enhancing effect of EGCG on MNP_cell_ was mediated by 67LR, co-treatment of a 67LR peptide (67LRp, amino acids 161–170 of 67LR) or knockdown of 67LR expression with esiRNA was performed. Figure 4A illustrates that 67LRp attenuated the enhancement effect of EGCG (10 μM) on MNP_cell_ by 45% in the Mag− group. In the Mag+ group, 67LRp attenuated the effect of EGCG at 5 and 10 μM by 38% and 19%, respectively (Figure 4B). Interestingly, the enhancement effect of EGCG on MNP_cell_ was almost completely blocked by esiRNA-mediated knockdown of 67LR in the absence of magnetic influence (Figure 4C), but not in the Mag+ group (Figure 4D). Expression of 67LR protein was reduced by 13% and 41% after transfection of 50 and 100 nM 67LR esiRNA, respectively (Figure 4D insert). Thus, 67LR appears to play a crucial role in the EGCG-enhanced MNP_cell_ only in the Mag− group.

### 3.3. Effect of Signaling Downstream of 67LR on EGCG-Induced MNP Uptake

To determine whether signal pathways downstream of 67LR may participate in the effect of EGCG on MNP uptake, LN-229 cells were pretreated with PI3K inhibitor (LY294002) or Akt inhibitor (Akti) for 30 min prior to 24 h incubation of dextran-MNP and EGCG. LY294002 at 10 and 30 μM significantly attenuated the effects of EGCG on MNP_cell_ in the absence (Mag−; Figure 5A) and presence (Mag+; Figure 5B) of the magnet. In addition, Akti also attenuated EGCG-enhanced MNP_cell_ with or without the magnet in a concentration-dependent manner (Figure 5C,D). These results suggest that the PI3K/Akt signaling pathway may be involved in the EGCG-enhanced MNP internalization.

Figure 6A illustrates that EGCG, C225 (an EGFR-blocking antibody), or EGF induced an increase in the phosphorylation of Akt pT308 but not Akt pS473 in LN-229 cells. EGCG induced a 4.8-fold Akt pT308, which is representative of two experiments with Akt pT308 increase by an average of 3.4-fold in response to EGCG. No effect of eNOS phosphorylation was observed in response to EGCG, C225, or EGF in this system. In contrast, EGCG induced Akt pS473 by 1.2-fold in A431 cells (Figure 6B). Figure 6C illustrates that C225 or EGF alone did not change the level of MNP_cell_, and neither did pretreatment with C225 or EGF prior to EGCG administration, suggesting that EGFR did not participate in the enhancement effects of EGCG on MNP_cell_. EGFR expression was relatively higher in A431 (Figure 6E) than in LN-229 (Figure 6D) cells, which was not altered by treatments of C225, EGF, or EGCG.

We then investigated whether the downstream molecule, cGMP, could regulate MNP uptake using pharmacological agents modulating sGMP levels in the cells. Pretreatment of 3 or 10 μM BAY41-2272 did not affect MNP_cell_, but increased EGCG-enhanced MNP_cell_ by 24% or 91% in the Mag− group (Appendix A). In the Mag+ group, BAY41-2272 did not alter MNP_cell_ with or without EGCG (Appendix A). NS-2028 did not alter the effects of EGCG or magnet on MNP_cell_ (Appendix A). Further studies indicate that 8-Br-cGMP (Appendix A) and zaprinast (Appendix A) exhibited no effects on the enhancement effects of EGCG, suggesting cGMP may not mediate the effects of EGCG.

Appendix A shows that 10 and 30 μM of desipramine attenuated EGCG-induced enhancement of MNP_cell_ by 8.9% and 39.3% in the Mag− group, respectively. In the Mag+ group, MNP_cell_ was enhanced up to 6.6-fold by EGCG compared with the control group, which was minorly attenuated by 5.7% and 11.2% by desipramine at 10 and 30 μM, respectively (Appendix A). The results suggest that acid sphingomyelinase may be involved in the effects of EGCG.

## 4. Discussion

To our knowledge, this study is the first demonstration of 67LR mediating/enhancing nanoparticle internalization induced by EGCG in the milieu. Following simple mixing after co-administration, EGCG may bind to MNPs and interact with 67LR on the cell surface to augment cellular uptake of nanoparticles. We demonstrate that EGCG-enhanced MNP uptake by LN-229 glioma cells was greatly reduced by blocking EGCG binding to 67LR or knocking down 67LR expression. The results suggest a crucial role of 67LR in EGCG-induced nanoparticle uptake.

Our results are consistent with previous studies that show that uptake of EGCG-conjugated nanocomposites is significantly higher than that without EGCG as a ligand [13,34,35]. In the current study, 67LR appears to mediate the enhancement effects of EGCG, which do not require immobilization of EGCG on the surface of nanoparticles. Furthermore, the presence of EGCG with MNPs in the culture medium appears to stabilize the aggregated needle structure of MNPs induced by 5-min magnet treatment, suggesting a direct interaction between EGCG and MNPs, probably via hydrophobic interaction. The results are in parallel with our previous finding that most of the antioxidant activity of EGCG is exhibited on the surface of dextran-coated MNPs after mixing EGCG with the nanoparticles in the culture medium [14]. Without the magnet, 67LR knockdown completely blocks EGCG-induced enhancement effects, suggesting that EGCG on MNPs may interact with 67LR and induce endocytosis of MNPs. Such enhanced aggregation of MNPs by EGCG and magnetic force may also be attributed to an increase in the vesicle size of internalized MNPs. However, the synergy of EGCG and magnetic force on MNP uptake was not affected by 67LR knockdown. It is likely that the capacity of MNP uptake induced by EGCG in the magnetic field is high enough to overcome 67LR knockdown.

The results from experiments with a 67LR-specific antibody, 67LR peptide, or heparin strongly suggest that 67LR mediates the effects of EGCG. Although it has been proposed that the laminin YIGSR peptide blocks the EGCG–67LR interaction, there was no effect on EGCG-enhanced MNP uptake by YIGSR peptide treatment. Since laminin binds to 67LR in a heparin-dependent manner [28], it is conceivable that the YIGSR peptide may not exert an effect in the experimental conditions without heparin.

In addition, the EGCG–67LR interaction may activate an PI3K/Akt pathway that subsequently modulates the endocytosis process; the downstream NO may accelerate endocytosis by S-nitrosylating dynamin, which increases dynamin oligomerization and GTPase activity, thereby cleaving the endocytic vesicle free from the plasma membrane [43,44]. In the current study, PI3K/Akt inhibitors attenuated cellular uptake of MNPs (Figure 5), which is in parallel with the previous finding that PI3K/Akt activation may participate in the endocytosis process of silver nanoparticles [45]. Inhibition of PI3K/Akt attenuated MNP uptake with or without EGCG, suggesting that PI3K/Akt-mediated endocytosis may be involved in MNP uptake. Although co-localization of eNOS and dynamin has been demonstrated by confocal microscopy and immunoprecipitation [46], and both dynamin-1 and Akt have been reported to participate in clathrin-mediated endocytosis [47], our study does not support a role of NO nor cGMP in the effect of EGCG-enhanced MNP uptake by LN-229 cells. Nevertheless, we cannot rule out the possibility that any of the pharmacological agents might exert an effect on the nano–bio interface that hinders appropriate interpretation of the effects of these agents.

The synergetic effect of EGCG and magnetic force on MNP uptake by glioma cells is consistent with previous studies [14,15,16], but not limited to glioma cells since A431 cells exhibited similar synergetic effects. It is implicated that magnetic force-induced MNP aggregation allows endocytosis machinery to internalize more particles using transmission electron microscopy and flow cytometry [48], without causing cytotoxicity [14,15] at the concentration of EGCG used in the current study. Nevertheless, the synergetic effects of magnetic force and EGCG resulted in a similar level of MNP_cell_, probably due to unsaturated capacity that internalizes most of the MNPs added to the culture, as described previously [49].

In the current study, C225, the antibody against EGFR activation, exhibited no effect on EGCG-induced MNP uptake (Figure 6C). Although previous study has demonstrated that EGCG may downregulate EGFR expression on the cell surface [40,42], EGCG did not exhibit an effect on EGFR phosphorylation nor expression in the current experimental conditions. Furthermore, A431 cells with much higher levels of EGFR expression did not exhibit enhanced MNP uptake in response to EGCG (Figure 1). Therefore, our study does not support a role of EGFR in the enhancement effects of EGCG on MNP uptake. However, our finding that EGCG induced Akt T308/S473 phosphorylation is consistent with Kumazoe et al. demonstrating Akt activation following EGCG–67LR interaction [24]. A previous study indicated that EGCG attenuated Akt pS473 to exert an anti-tumor effect [50]. The mechanism of the discrepancy is unknown, but it may be due to different treatment models. Since PI3 kinase-dependent Akt pT308 may be mediated by other kinases downstream of PI3 kinase [51], EGCG-induced Akt pT308 may involve a direct or indirect effect of PI3 kinase.

Recently, the galloyl moiety of EGCG has been reported to antagonize transforming growth factor-β (TGF-β)-mediated cell invasion and metastasis by interaction within the kinase domain of the TGF-βR1 [52]. We cannot rule out the possibility of other EGCG-binding proteins in the enhancement effects of EGCG, such as the intermediate filament protein vimentin, which is essential in maintaining the structure, mechanical integration, and function of cells [53].

## 5. Conclusions

Our study indicates that the enhancement effect of EGCG on nanoparticle uptake may be mediated by 67LR binding and subsequent signaling such as Akt phosphorylation. The enhancement effect of EGCG on MNP_cell_ is synergistically augmented by the application of a magnetic field, which may be due, at least in part, to an enhancement effect of EGCG on particle–particle interaction. The interplay of EGCG and magnetic force may be applied in vitro, and potentially in vivo, to enhance internalization of MNPs by tumor cells with 67LR expression.

## Figures and Tables

**Figure 1 pharmaceutics-14-01523-f001:**
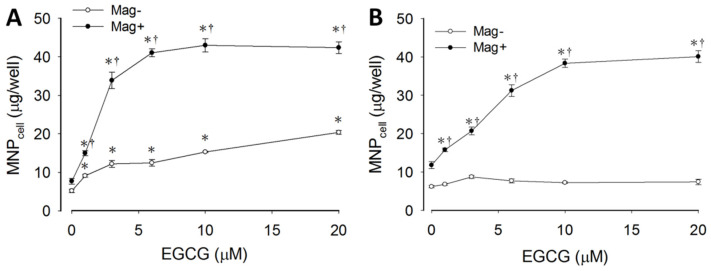
EGCG-enhanced uptake of MNPs by tumor cells. LN-229 (**A**) and A431 (**B**) cells were incubated with CMX-MNPs (50 μg/well) in the absence (Mag−) or presence (Mag+) of the magnet for 2 h prior to determination of cell-associated MNPs (MNP_cell_). Values are mean ± SEM (*n* = 4). *^, †^
*p* < 0.05 compared with corresponding group without EGCG or magnet, respectively.

**Figure 2 pharmaceutics-14-01523-f002:**
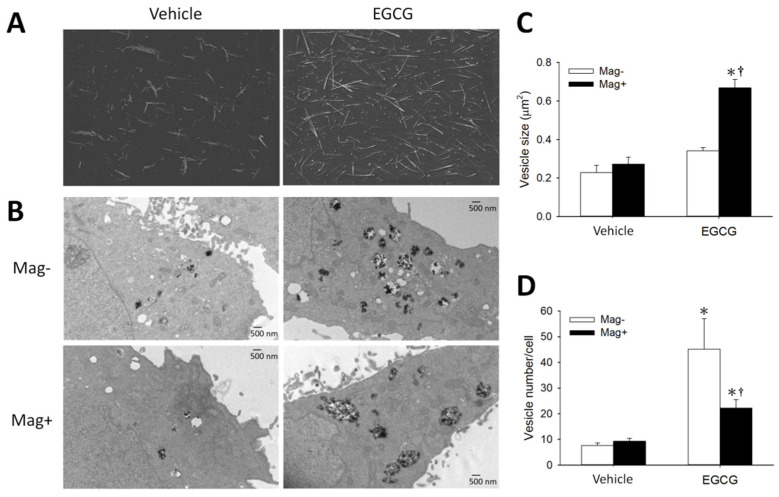
EGCG-enhanced MNP aggregation and internalization by glioma cells. (**A**) MNP aggregates in cell-free medium were observed by 3D Cell Explorer with or without EGCG for 2 h. (**B**) U87MG cells were incubated with dextran-MNPs (100 μg/mL; 25 μg/cm^2^) and EGCG (10 μM) in the absence (Mag−) or presence (Mag+) of the magnet for 24 h, followed by analysis with transmission electron microscopy. The vesicle size (**C**) and vesicle number (**D**) were analyzed from 6–8 cells in each group with (Mag+) or without (Mag−) magnetic influence. Values are mean ± SEM (*n* = 56–226). *^, †^
*p* < 0.05 compared with corresponding group without EGCG or magnetic field, respectively.

**Figure 3 pharmaceutics-14-01523-f003:**
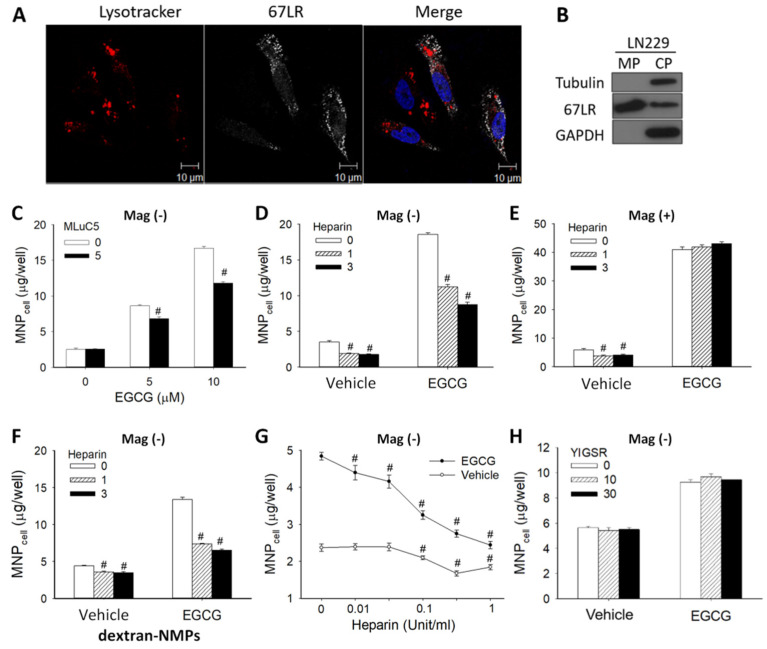
Anti-67LR antibody and heparin, but not YIGSR, blocked EGCG-induced MNP uptake in the absence of the magnet. (**A**) Lysosomes (red), 67LR (white), and nuclei (blue) of LN-229 cells were stained and observed with confocal microscopy. (**B**) Expressions of tubulin, 67LR, and GAPDH in the fractions of membrane (MP, 0.6 μg protein) and cytoplasm (CP, 20 μg protein) were determined by western blot. After pretreatment with anti-67LR antibody (MLuC5, 5 μg/mL; (**C**)), heparin (0.01–3 Unit/mL; (**D**–**G**)) or YIGSR (10 or 30 μg/mL; (**H**)), LN-229 cells were incubated with 50 μg/well of CMX-MNPs (**C**–**E**,**G**,**H**) or dextran-MNPs (**F**) in the presence of EGCG (5 or 10 μM) or vehicle for 2 and 6 h, respectively. The incubation was conducted in the absence (Mag−; (**C**,**D**,**F**–**H**)) or presence (Mag+; (**E**)) of the magnet. The results are representative of 4 experiments with different batches of cells; values are presented as mean ± SEM (*n* = 4). ^#^
*p* < 0.05 compared with the corresponding group without MLuC5 or heparin.

**Figure 4 pharmaceutics-14-01523-f004:**
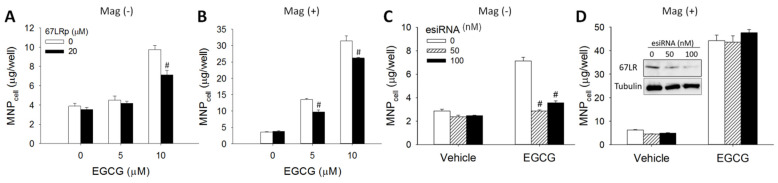
Peptide of 67LR (67LRp) and 67LR knockdown attenuated the effects of EGCG on MNP uptake by LN-229 cells. (**A**,**B**) LN-229 cells were incubated with 50 μg/well of dextran-MNPs, EGCG (5 or 10 μM), and 67LRp (20 μM) for 6 h. (**C**,**D**) After transfection with esiRNA (50 or 100 nM), LN-229 cells were incubated with 50 μg/well of CMX-MNPs and EGCG (10 μM) for 2 h. The incubation was conducted in the absence (Mag−; (**A**,**C**)) or presence (Mag+; (**B**,**D**)) of the magnet. The 67LR expression was determined 96 h after transfection with esiRNA using western blotting ((**D**) insert). Values are mean ± SEM (*n* = 4). ^#^
*p* < 0.05 compared with the corresponding control group.

**Figure 5 pharmaceutics-14-01523-f005:**
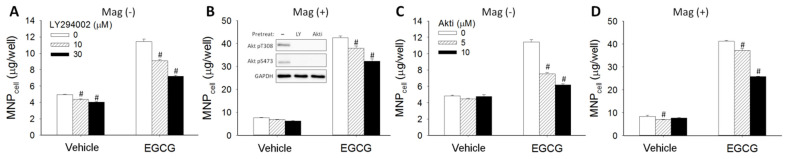
PI3K/Akt inhibitor attenuated EGCG-induced MNP uptake. After 30-min pretreatment with LY294002 (PI3K inhibitor; 10 or 30 μM; (**A**,**B**)) or Akti (Akt inhibitor; 5 or 10 μM; (**C**,**D**)), LN-229 cells were incubated with dextran-MNPs (50 μg/well) with EGCG (10 μM) in the absence (**A**,**C**) or presence (**B**,**D**) of the magnet for 6 h. Cells were pretreated with LY294002 (30 μM) or Akti (10 μM) for 30 min and treated with EGCG (25 μM) for 1 h. Phosphorylation of Akt at Ser473 and Thr308 was measured by western blotting (B insert). Values are mean ± SEM (*n* = 4); the results are representative of 4 experiments using different batches of cells. ^#^
*p* < 0.05 compared with the corresponding control group.

**Figure 6 pharmaceutics-14-01523-f006:**
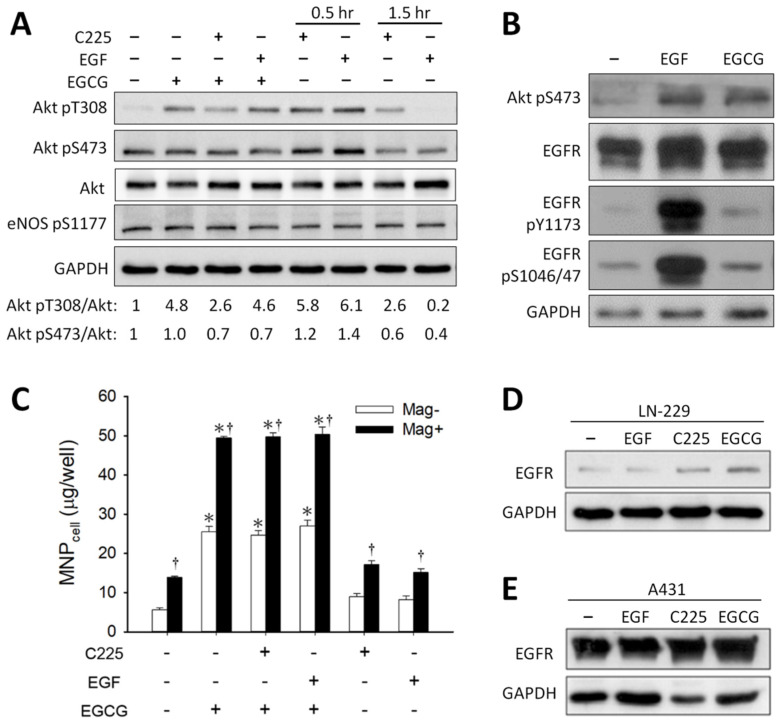
EGCG- and EGF-induced Akt phosphorylation of glioma cells. Western blot analysis was conducted in (**A**) LN-229 cells pretreated with C225 (5 μg/mL) or EGF (50 ng/mL) for 30 min prior to exposure of the cells to EGCG (25 μM) for 1 h; (**B**) A431 cells were treated with EGF (50 ng/mL) or EGCG (25 μM) for 15 min. (**C**) After pretreatment with C225 or EGF for 1 h, LN-229 cells were incubated with MNPs (CMX-MNP; 50 μg/well) and EGCG in the absence (Mag−) or presence (Mag+) of the magnet for 4 h prior to determination of cell-associated MNPs (MNP_cell_). The results are representative of three experiments with different batches of cells; values of MNP_cell_ are presented as mean ± SEM (*n* = 4). ^*, †^
*p* < 0.05 compared with corresponding group without EGCG or magnetic field, respectively. Relative expression of total EGFR levels was determined in LN-229 (**D**) or A431 (**E**) cells.

## Data Availability

Not applicable.

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
