# Peer review of "Laminin Receptor-Mediated Nanoparticle Uptake by Tumor Cells: Interplay of Epigallocatechin Gallate and Magnetic Force at Nano–Bio Interface"

_pharmaceutics, 2022, doi:10.3390/pharmaceutics14081523_

Round 1

Reviewer 1 Report

In this article, Hsu et al, describe 67-KDa Laminin receptor (67LR) as one of the membrane components responsible for the uptake of epigallocatechin gallate (EGCG) magnetic nanoparticles. The authors provide organized and clear data that suggests EGCG is internalized via 67LR. This research article aligns with the aim and scope of the Pharmaceutics journal. We suggest the authors address the following concerns so we can consider publishing their work in the Journal.

1.- Some sentences are missing verbs, for example, in lines 44-46, the second sentence is incomplete. Similar mistakes are found in the paragraph where figure 6 is described.

2. More detailed information about the magnet used and the ‘homemade plate” is needed. Please, include the magnetic properties of the magnet used.

3. There seems to be a particle uptake limit when using the magnet. Please address if the limit is biological, or if it is related to the magnetic properties. If pertinent discuss the influences of the magnetic properties’ strength in the uptake of the particles.

4. Please provide a rationale for why LN-229, U87MG, and A431 cell lines were used.

5.- Comparison between graphs is hard since the Y axes in the figures are not consistent, some of them go to 20 MNP while some go to 50 MPN.

6.- Figures 3, C-H are hard to interpret. Figures 3D and 3E can be presented in the same graph, that way the influence of the magnet on the particle uptake is evident.

7.- The impact of 1 unit of Heparin in the vehicle and EGCG groups is different between graphs from figures 3D and 3G, especially the heparin 0/EGCG group, D graphs show a higher uptake than graph G. Please explain why this difference is observed.

8.- Please include the influence of heparin, in A431 cell line uptake.

9.- In the introduction, an extensive list of biological effects of catechins is included. As of now, the paper shows a possible increase in cellular uptake of particles. Since the cell lines used are cancer-derived, the authors should consider adding information on dose-dependent pro-apoptotic effects.

Author Response

Reviewer #1:

1. Some sentences are missing verbs, for example, in lines 44-46, the second sentence is incomplete. Similar mistakes are found in the paragraph where figure 6 is described.

A- In the revised version, the sentence is modified as follows,” Subsequently, the interaction between drug-loaded nanoparticles and tumor cells is often the key to cellular internalization that may be crucial to the efficacy of chemotherapeutic drugs [5-7].” The legend of Figure 6 has been modified as following,”….. (B) A431 cells were treated with EGF (50 ng/ml) or EGCG (25 mM) for 15 min.”

2. More detailed information about the magnet used and the ‘homemade plate” is needed. Please, include the magnetic properties of the magnet used.

A- More detailed information was provided as following,” As described previously [14], culture cells in 24-well plates were subjected to magnetic force by placement on top of a homemade plate filled with cylindrical NdFeB magnets with a magnetic field of 3.4 kGauss at the center of each well to facilitate MNP sedimentation.....”

3. There seems to be a particle uptake limit when using the magnet. Please address if the limit is biological, or if it is related to the magnetic properties. If pertinent discuss the influences of the magnetic properties’ strength in the uptake of the particles.

A- Tumor cells exhibit big capacity for internalization of MNPs, which was characterized in a previous study (Lu YC et al., JMMM 427:71-80, 2017). Reach of uptake limit in the presence of the magnet was simply due to consumption of more than 80% of MNPs added to the culture plate. Therefore, the end of the 5th paragraph was modified as following, “Nevertheless, the synergetic effects of magnetic force and EGCG resulted in a similar level of MNPcell, probably due to unsaturated capacity that internalize most of MNPs added to the culture, as described previously [51].”

4. Please provide a rationale for why LN-229, U87MG, and A431 cell lines were used.

A- Human glioma cells, LN-229 and U87MG, were used because the effects of EGCG were first characterized in these cells (Lu YC et al., Nanoscale 6:10297, 2014), as explained in the 1st sentence of the 3rd paragraph in Introduction. A431 cells with high EGFR expression were used in some experiments to determine whether the effects of EGCG was mediated by interaction with EGFR. In the revised version, the 1st sentence of the 5th paragraph in Discussion was modified as following, “The synergetic effect of EGCG and magnetic force on MNP uptake by glioma cells are consistent with previous studies [14-16], but not limited to glioma cells since A431 cells exhibited similar synergetic effects.” In addition, a sentence in the 6th paragraph of Discussion is modified as following, “…..A431 cells with a much higher levels of EGFR expression did not exhibit enhanced MNP uptake in responses to EGCG (Figure 1). Therefore, our study does not support a role of EGFR in the enhancement effects of EGCG on MNP uptake.”.

5. Comparison between graphs is hard since the Y axes in the figures are not consistent, some of them go to 20 MNP while some go to 50 MPN.

A- We agree with the reviewer that a consistent scale of the Y axes may be more convenient in comparison across graphs.  However, the synergetic effects of EGCG and magnetic force have been very reproducible, as demonstrated in Figure 1. Therefore, different scale may reveal more details of the change in MNP uptake, especially in the absence of the magnet (Mag-).

6. Figures 3, C-H are hard to interpret. Figures 3D and 3E can be presented in the same graph, that way the influence of the magnet on the particle uptake is evident.

A- We have slightly modified the labeling in the Figure 3 C-H to help reading. Figure 3D and 3E demonstrates that heparin may exhibit an effect in Mag-, but not Mag+ group; whereas Figure 3D and 3F demonstrates suppressing effect of heparin on EGCG-induced uptake of CMX-MNPs and dextran-MNPs in Meg-, respectively. Therefore, it is necessary to present Figure 3D and 3E in a separate manner.

7. The impact of 1 unit of Heparin in the vehicle and EGCG groups is different between graphs from figures 3D and 3G, especially the heparin 0/EGCG group, D graphs show a higher uptake than graph G. Please explain why this difference is observed.

A- The background of MNPs uptake in the absence/ presence of EGCG are variable, which may be due to different batch of MNPs or cells.

8. Please include the influence of heparin, in A431 cell line uptake.

A- We used different molecular and pharmaceutical approaches to examine 67LR-mediated MNP uptake induced by EGCG in LN-229 cells. A431 cells were not used to support the experimental results from LN-229 cells because EGCG per se did not induce MNP uptake in the absence of the magnet (Meg-), as illustrated in Figure 1. The discrepancy of the effects of EGCG under Mag- in LN-229 vs. A431 is uncertain. It is possible that there may be more EGCG-binding sites, such as EGFR, on the surface of A431 cells that may compromise the effect of EGCG via 67LR.

9. In the introduction, an extensive list of biological effects of catechins is included. As of now, the paper shows a possible increase in cellular uptake of particles. Since the cell lines used are cancer-derived, the authors should consider adding information on dose-dependent pro-apoptotic effects.

A- We agree that EGCG may induce apoptosis in cancer cells, as reported in the literature. The pro-apoptosis effect of EGCG was also addressed in our previous studies (Lu et al., 2014; ref #14; Cheng et al., 2019 ref #15). No cell viability change was observed even at 30 mM EGCG treatment on LN-229 cells.  The 5th paragraph in Discussion is modified as following, “It is implicated that magnetic force-induced MNP aggregation allows endocytosis machinery to internalize more particles using transmission electron microscopy and flow cytometry [48], without causing cytotoxicity [14-15] at a concentration of EGCG as in the current study.”

Reviewer 2 Report

see attached

Author Response

Reviewer #2:

1) title of the article does not reflect the main focus on EGCG, and should be revised

A- The title has been modified as suggested, “Laminin Receptor-Mediated Nanoparticle Uptake by Tumor Cells: Interplay of Epigallocatechin Gallate and Magnetic Force at Nano-Bio Interface”.

2) results section should be re-structured and subdivided using specific subheadings describing the major finding of the paragraph

A- We agree that subheading may improve comprehension. Three subheadings have been used to structure the result section in the revised version, i.e. 1) synergetic effects of EGCG and magnetic force on MNP uptake, 2) 67LR-mediated MNP uptake induced by EGCG, and 3) signaling downstream of 67LR on EGCG-induced MNP uptake.

3) discussion should be revised, especially concerning the inclusion of relevant previous work, eg. recent study of Mu et al. describing an EGCG nanoreactor for glioblastoma therapy [1]; reports on effect on cancer cells, also mediated by laminin! [2,3], and several more.

A- We agree with the reviewer to include more references. The 5th paragraph of Introduction is modified as following, “Such effects may facilitate intracellular delivery of nanocarriers with therapeutic agents. Nanoparticles with immobilized EGCG have been shown to exhibit enhanced uptake by tumor cells [34-35] or enhanced therapeutic efficacy in tumor models [13, 34-37]; limited evidence indicated that 67LR antibody attenuates the enhancement ef-fects of EGCG on cellular uptake of EGCG nanocompositeset [34,36]. However, it re-mains unknown whether a receptor on the plasma membrane may mediate the en-hancing effect of EGCG in the milieu on cellular uptake of nanoparticles.” In addition, the 2nd paragraph of Discussion is also modified as following, “Our results are consistent with previous studies that uptake of EGCG-conjugated nanocomposites are significantly higher than that without EGCG as a ligand [13] add #43, Chavva et al., Nanomaterials 2019; Mu et al., J Adv Res, 2021.”  However, suggested reference #3 (Zeng, L et al. (Sci Rep 2017, 7, 45521) was not included due to lack of reproducible uptake results, with only representative images from confocal microscopy.

4) C225 should be explained in the results section at first time mentioned - results of eNOS activation (blot Figure 6A) are not mentioned in the text. There seems to be no effect of all tested compound/conditions? - authors should explain why cGMP

A- We agree that results of eNOS phosphorylation should be mentioned, which is consistent with the findings that alteration of cGMP levels did not exert an effect on MNP uptake. The following sentence is added in the text of Results in the revised version,” Figure 6A illustrates that EGCG, C225 (an EGFR blocking antibody) or EGF induced an increase in …..with Akt pT308 increase by an average of 3.4-fold in response to EGCG. No effect of eNOS phosphorylation was observed in response to EGCG, C225 or EGF in this system.”

5) modulating agents were analyzed, is there already evidence that cGMP plays a role in MNPs uptake? - use of # in figures for significance? Established presentation of *,**, and so on should be used

A- There is no evidence demonstrating a role of cGMP in MNP uptake; however, interaction of EGCG and 67LR may stimulate an increase in cGMP levels. The 5th paragraph regarding 67LR signaling is modified as following, “…..the downstream signaling pathways of EGCG-67LR have been demonstrated to alter the structure/function of lipid rafts and thus modulate the endocytosis process; interaction of EGCG and 67LR induced activation of protein kinase B (PKB; Akt)/ endotheli-al nitric oxide synthase (eNOS)/ nitric oxide (NO)/ soluble guanylyl cyclase (sGC)/ cyclic guanosine monophosphate (cGMP) pathway that has been shown to increase membrane fluidity…..”. However, use of # for significance in the figures is widely accepted by different journals, including Nanoscale, Throm Haemost. Beilstein J Nanotechnol, Intl J Nanomed etc.

6) some minor spelling (eg. manget ), preposition errors, eg. heading of figure 6 akt phosphorylation of glioma cells really sense; p.9, l. 341

A- Minor spelling errors have been corrected in the revised version.

7) Unfortunately, supplementary video was not available.

A- Supplementary video is uploaded as a separated file in revision.

Round 2

Reviewer 1 Report

The comments were addressed